# Modular Assembly of Phosphite Dehydrogenase and Phenylacetone Monooxygenase for Tuning Cofactor Regeneration

**DOI:** 10.3390/biom11060905

**Published:** 2021-06-17

**Authors:** Ni Nyoman Purwani, Caterina Martin, Simone Savino, Marco W. Fraaije

**Affiliations:** Molecular Enzymology Group, University of Groningen, Nijenborgh 4, 9747 AG Groningen, The Netherlands; n.n.purwani@rug.nl (N.N.P.); c.martin@rug.nl (C.M.); s.savino@rug.nl (S.S.)

**Keywords:** oligomers, self-assembly, RIAD–RIDD tag, Baeyer–Villiger monooxygenase, cofactor regeneration

## Abstract

The use of multienzyme complexes can facilitate biocatalytic cascade reactions by employing fusion enzymes or protein tags. In this study, we explored the use of recently developed peptide tags that promote complex formation of the targeted proteins: the dimerization-docking and anchoring domain (RIDD–RIAD) system. These peptides allow self-assembly based on specific protein–protein interactions between both peptides and allow tuning of the ratio of the targeted enzymes as the RIAD peptide binds to two RIDD peptides. Each of these tags were added to the C-terminus of a NADPH-dependent Baeyer–Villiger monooxygenase (phenylacetone monooxygenase, PAMO) and a NADPH-regenerating enzyme (phosphite dehydrogenase, PTDH). Several RIDD/RIAD-tagged PAMO and PTDH variants were successfully overproduced in *E. coli* and subsequently purified. Complementary tagged enzymes were mixed and analyzed for their oligomeric state, stability, and activity. Complexes were formed in the case of some specific combinations (PAMO_RIAD_–PTDH_RIDD_ and PAMO_RIAD/RIAD_–PTDH_RIDD_). These enzyme complexes displayed similar catalytic activity when compared with the PTDH–PAMO fusion enzyme. The thermostability of PAMO in these complexes was retained while PTDH displayed somewhat lower thermostability. Evaluation of the biocatalytic performance by conducting conversions revealed that with a self-assembled PAMO–PTDH complex less PTDH was required for the same performance when compared with the PTDH–PAMO fusion enzyme.

## 1. Introduction

In nature, metabolic pathways are often catalyzed by multienzyme complexes. Such complexes are efficient in catalyzing cascade reactions and prevent accumulation of (unwanted) reaction intermediates [1,2]. The use of enzyme complexes offers advantages over conventional step-by-step reactions used in biocatalysis, such as acceleration of the overall reaction rate, elimination of product inhibition, efficient transfer of intermediates, and higher stability. All these elements are reflected in an overall higher catalytic performance [3,4,5,6]. A well-known example of a two-enzyme cascade reaction setup in biocatalysis is the use of an enzyme for cofactor regeneration when dealing with a cofactor-dependent enzyme [7,8]. In recent years, the use of fusion enzymes has been developed for this by fusing the respective genes encoding these enzymes [7,8]. Fusing two or more enzymes enables production of the enzymes in one step while it also can result in improved stability and catalytic performance compared with separate enzymes. However, it has also been observed that fusing enzymes can lead to problems in protein production, stability, and activity [8].

Baeyer–Villiger monooxygenases (BVMOs) have been shown to be valuable biocatalysts [9]. Most BVMOs are FAD-containing enzymes that depend on NADPH for activity. To facilitate regeneration of NADPH, we have developed an expression system for producing BVMOs fused to a stable variant of phosphite dehydrogenase [8,10]. Various BVMOs could be produced and used as bifunctional fusion enzymes, including cyclohexanone monooxygenase (CHMO), phenylacetone monooxygenase (PAMO), and cyclopentanone monooxygenase (CPMO). The bifunctional enzymes were found to retain roughly the same kinetic properties and chemo-, regio-, and stereoselectivity. A slight increase in *K*_M_ values for NADP^+^ and phosphite was observed for phosphite dehydrogenase (PTDH) when fused to a target enzyme. Recently, we have also reported on the fusion of three distinct alcohol dehydrogenases (ADHs) with CHMO for conversion of cyclohexanol to caprolactone in a cascade reaction [11]. Two of the tested ADHs were found to be inactive when produced as N-terminal fusions (ADH–CHMO). This shows that fusions cannot always be produced or suffer from poor stability due to structural incompatibility. Nevertheless, the successfully produced ADH–CHMO efficiently converted cyclohexanol (99% conversion), compared with only 41% conversion when the two enzymes were used as separately produced and purified protein.

Despite the advantages of producing fusion enzymes, there are some drawbacks that may result from fusing enzymes, such as problems in heterologous production, loss of enzyme activity, and the limitation of only a one-to-one enzyme ratio. In this study we generated enzyme complexes of PTDH and PAMO (Figure 1A,B) by equipping each enzyme with a tag that allows self-assembly based on a dock-and-lock mechanism. We exploited the peptide–peptide interactions occurring between the regulatory subunit of protein kinase and the A-kinase anchor protein. The respective domains are individually referred to as the dimerization docking domain (RIDD, 44 residues) and anchor domain (RIAD, 18 residues), respectively (Figure 1C). The RIAD and RIDD peptides form tight complexes through various noncovalent protein–protein interactions [12,13,14,15]. Two RIDD peptides form a stable dimeric α-helical docking domain that has a tight interaction with the RIAD anchor peptide (Figure 1). It has been shown that these peptides can be used as protein tags to force formation of protein complexes [16,17]. Each peptide can be added as a tag to the N- or C-terminus of a target protein using a linker with optimized length and amino acid composition. Two or more RIAD sequence repeats can be used in tags to facilitate scaffold formation for binding of the RIDD module. An advantage of this approach is the feature that any tagged protein can be produced and stored independently and can be combined for several types of RIDD–RIAD combinations. It also allows tuning of the ratio of the complexed proteins. It has been employed to generate bispecific trivalent complexes comprising three antibody fragments [18]. Inspired by previous results, another construct was later developed by generating an IgG–RIAD module combined with the same RIDD-module, producing hexavalent antibodies [19]. The RIAD/RIDD system has also been used for the construction of immunocytokines from various cytokines, including interferon alpha (IFNα), erythropoietin, and granulocyte colony-stimulating factor (G-CSF) [20,21]. Recently, Kang and co-workers [22] used RIAD/RIDD-mediated multi-enzyme complex formation for the in vivo assembly of enzymes which enhanced the metabolic flux for carotenoid production. This demonstrates that formation of multi-enzyme complexes can promote biocatalytic cascade reactions. In this study, we have specifically tagged PTDH and PAMO with RIAD and RIDD tags on their C-termini (see Figure 1) in order to generate various PTDH–PAMO complexes. After complex formation, the biocatalysts were evaluated in comparison with the PTDH–PAMO fusion enzyme and the non-fused enzymes.

## 2. Materials and Methods

### 2.1. Gene Cloning

The DNA fragments encoding the RIAD, RIAD–RIAD, and RIDD tags were designed using online software (Benchling, San Fransisco, CA, United States). Recognition sites for the restriction enzyme BsaI (GGTCTCN) at the beginning and (NGAGACC) at the end of the coding sequences were introduced to allow Golden Gate cloning. Detailed information of the designed gene sequences is in the Appendix A. The plasmid used for amplification of the PTDH- and PAMO-encoding genes was the pBAD-based pCRE2-PAMO [23]. BsaI restriction sites already present in the PAMO sequence were mutated using PfuUltra II Hotstart PCR Master Mix (Agilent, Santa Clara, CA, United States) to avoid unwanted restriction activity, but maintaining the same translated amino acid. The PTDH and PAMO genes were amplified using the primers reported in Appendix A. Fusions of synthetic DNA encoding the RIAD, RIAD–RIAD, and RIDD tags to the C-termini of PTDH and PAMO were obtained by inserting the linker GGGGS in between. The six new fusions constructs were obtained through Golden Gate cloning using a pBAD vector with a 6× His-tag positioned at the N-terminus. The correct sequences were confirmed by sequencing. The recombinant enzymes produced are named as PAMO_A_, PAMO_A2_ PAMO_I_, PTDH_A_, PTDH_A2_, PTDH_I_ (where A indicates the presence of an RIAD tag, and I indicates the presence of the RIDD tag).

### 2.2. Heterologous Production and Purification of Fusion Enzymes

Production and purification of the six recombinant proteins were performed as previously described [23]. Cells were grown in Terrific Broth (TB) and when the OD_600_ reached 0.7–0.8, protein production was induced by addition of arabinose (final concentration 0.02%) for all variants. Protein production was carried out overnight at 24 °C. The fusion enzymes were purified using Ni Sepharose resin (GE Healthcare, Bio-Sciences AB, Chicago, IL, United States). Purified enzymes (in 50 mM Tris–HCl, pH 7.5) were flash frozen in liquid nitrogen and stored at −80 °C.

### 2.3. Formation of Multi-Enzyme Complexes

Enzyme complexes were prepared by mixing 1.5 mg of RIAD-tagged enzyme (20 µM) with twice the molar excess of RIDD-tagged enzyme in 50 mM Tris–HCl buffer with 150 mM NaCl, 0.02% Tween-20 and 1.0 mM EDTA, pH 7.5. When mixing RIAD–RIAD-tagged proteins, the ratio of enzymes was 1:4 (RIAD–RIAD:RIDD). The mixtures were analyzed by size exclusion chromatography using a Superdex 200 (10/300) column on an AKTA Prime with a flow rate of 0.25 mL min^−1^ (50 mM Tris–HCl with 150 mM NaCl, pH 7.5). Elution volumes were used to determine the molecular size of observed protein fraction. Fractions containing enzyme complexes were pooled and concentrated for subsequent analysis by SDS–PAGE (pre cast gel, BioRad, Hercules, California, United Stated), and by ThermoFAD and ThermoFluor using a PCR machine (BioRad CFX96 touch real time, Hercules, CA, United Stated).

### 2.4. Thermostability Assay

For all purified enzymes, the apparent melting temperatures (T_m_) were determined by employing ThermoFAD (PAMO) or ThermoFluor (PTDH) [24]. The protein concentration for analysis was standardized to 10 μM (ThermoFAD) or 1.0 mg/mL (ThermoFluor). Samples were transferred to 96-well plates. The temperature was increased by 0.5 °C, every 30 s, from 25 to 90 or 99 °C in an RT-PCR thermocycler. ThermoFAD detects the unfolding of enzymes based on the release of the flavin cofactor. For ThermoFluor, SYPRO Orange dye was added to the sample which reports on protein unfolding.

### 2.5. Determination of Enzyme Activity

The activity of the purified enzymes was determined by monitoring spectrophotometrically the increase (PTDH) and decrease (PAMO) of NADPH absorbance at 340 nm over time. The reaction mixtures (100 µL) typically contained, 0.04–0.5 µM enzyme, substrate (1.0 mM phenylacetone or 5.0 mM Na_2_HPO_3_), 100 µM cofactor (NADPH or NADP^+^), 1 % (*v*/*v*) DMSO, and were incubated at 25 °C for 1 min. Kinetic parameters were obtained as described previously [23].

### 2.6. Biotransformation

Biotransformation using purified enzyme complexes was performed in 2 mL vials. The total reaction volume was 500 µL containing Tris–HCl buffer (50 mM, pH 7.5), 150 µM NADPH, 10 µM FAD, 10 mM Na_2_HPO_3_, and 5.0 µM PTDH–PAMO or 5.0 µM of a tagged PAMO variant with the corresponding concentration of tagged PTDH. Racemic bicyclo[3.2.0]hept-2-en-6-one (5.0 mM) was used as substrate. The mixture was subsequently shaken at 150 rpm, 24 °C for 5 h and 24 h. The mixture was extracted twice with 500 µL ethyl acetate (which included 0.025 mM as internal standard methyl ester of benzoic acid) for 60 s. Anhydrous magnesium sulphate was added to remove residual water. Analysis was carried out by gas chromatography (GCMS-QP2010, Ultra, Shimadzu, Kyoto, Japan) with electron ionization and quadrupole separation on a HP-1 column as previously described [25].

## 3. Results and Discussion

### 3.1. Enzymes Production

PTDH (dimeric) is an attractive biocatalyst for the regeneration of NADH or NADPH. Previously, we have shown that PTDH can be used as fusion protein to equip NADPH-dependent enzymes with a cofactor regenerating enzyme. Such enzyme fusions perform well as self-sufficient biocatalysts [26]. In the present study, we explored an alternative approach to fuse PTDH in a non-covalent manner with a target enzyme. We choose PAMO (monomeric) as a test enzyme as it is one of the best studied BVMOs and was shown to be a potent biocatalyst. The goal of this study was to obtain PTDH–PAMO complexes using the RIDD/RIAD tags. An appealing feature of the RIDD/RIAD system is that complex formation is not equimolar. Two RIDD-tagged protomers (as dimer) will bind to one RIAD-tagged protomer (see Figure 1). This allows tuning the ratio of different biocatalysts in the complexes by which one can compensate for differences in catalytic performance of each biocatalyst. For the generation of different complexes, three tagged PAMO and PTDH variants were produced in which an RIAD, an RIAD–RIAD, or an RIDD tag was fused to the C-terminus of each enzyme (Table 1). The tags were connected via a glycine-rich flexible linker while all proteins carried an N-terminal His-tag. All six tagged proteins were successfully produced and purified as soluble proteins (Appendix A). The tagged variants of PAMO were purified as yellow proteins, which confirms that they were still able to bind the FAD cofactor. The efficiency of FAD binding was determined by measuring and comparing the absorption at 441 nm (FAD) and 280 nm (protein) [27]. The A_280_/A_441_ ratio for the PAMO fusions was found to be around 14, indicating that PAMO was mainly in its active holo form.

### 3.2. Formation of Enzyme Complexes

The purified tagged proteins were analyzed for their oligomerization behavior. First, each purified protein was analyzed by size exclusion chromatography in the absence of any other tagged protein. As expected, analysis of PAMO_A_ and PAMO_A2_ (carrying one and two C-terminal RIAD tags, respectively) revealed an elution volume corresponding to a monomer (Appendix A). PAMO_I_ was found to elute as a homodimer which is also in agreement with the RIDD tag which forms dimers (Appendix A). All tagged PTDH variants were found to be dimeric. This is in line with PTDH forming stable dimers. Apparently, the RIDD-tagged PTDH is not able to form larger complexes (Appendix A).

Next, RIDD- and RIAD-tagged proteins were mixed to see whether RIDD–RIAD mediated complexes are formed. The tested mixtures were: PTDH_A_–PAMO_I_, PTDH_A2_–PAMO_I_, PAMO_A_–PTDH_I_, and PAMO_A2_–PTDH_I_. Complex formation was analyzed using size exclusion chromatography followed by SDS-PAGE analysis of specific elution fractions. Each fusion enzyme was mixed with the ratio described in the experimental part, according to the expected complex formation. The PTDH_A_–PAMO_I_ and PTDH_A2_–PAMO_I_ combinations successfully assembled into larger oligomers. The expected molecular weight for the assembly of PTDH_A_–PAMO_I_ was 364 kDa (dimeric PTDH_A_ (2 × 42 kDa) complexed to four PAMO_I_ protomers (4 × 70 Da)). The elution volume of 11.5 mL of the main peak observed upon size exclusion chromatography corresponds with 250 kDa (Figure 2A). The difference between the expected size of the oligomer and the size obtained from size exclusion chromatography may be due to the special hydrodynamic behavior of the oligomer. A different elution pattern was observed for the mixture of PTDH_A2_ and PAMO_I_. It was expected that dimeric PTDH_A2_, with in total four RIAD tags, could bind eight PAMO_I_ protomers, corresponding to a multimer of 650 kDa. Indeed, a peak at 9.25 mL is observed that corresponds to a protein 650 kDa (Figure 2B). Fractions obtained from this peak were collected and analyzed by SDS-PAGE, revealing the presence of two proteins: 45 kDa (PTDH_A2_) and 70 kDa (PAMO_I_) (Appendix A). This confirms that a PTDH_A2_–PAMO_I_ complex was formed. However, protein peaks with larger elution volumes were also observed. The main peak had an elution volume of 11.0 mL and corresponds to a complex with half of the size (310 kDa) than the expected PTDH_A2_–PAMO_I_ assembly. This could be due to the formation of an assembly with only four PAMO protomers bound to PTDH_A2_. Two minor peaks at 12.5 mL and 14.8 mL can be explained by some PTDH_A2_ and PAMO_I_, respectively, that did not form a complex. This can hint to a poor affinity for the tags to binds but can also be (partly) explained by an inaccurate estimation of protein amounts, resulting in an imperfect ratio of proteins for complex formation. Nevertheless, it was gratifying to observe that RIAD-tagged PTDH could assemble into larger oligomers with RIDD-tagged PAMO. 

Assembly of the other two complexes, PAMO_A_–PTDH_I_ and PAMO_A2_–PTDH_I,_ did not occur (Appendix A), since the observed elution volumes were quite similar compared with the elution of the individual enzymes. After mixing, both tagged enzymes co-eluted during elution in size exclusion chromatography. To determine the possible influence of the tags on assembly formation, other constructs were generated by using variants of PAMO only. Formation of a PAMO_A_–PAMO_I_ complex was successfully obtained (Appendix A) as evidenced by a main peak at 11.45 mL, corresponding to a size of 263 kDa (expected 207 kDa). This difference may be due to the special hydrodynamic behavior of the oligomer. The main peak of the PAMO_A2_–PAMO_I_ mixture eluted at 12.7 mL, corresponding to a size of 140 kDa while a size of 347 kDa (Appendix A) would be expected for a full complex formation (the expected oligomer was obtained as a minor peak indicating only a small portion of the enzymes–tag fusion worked for oligomer formation). Similar to the results with PTDH_A2_ mentioned above, the repeat of the RIAD tag seems to be incompatible with formation of a multimer. On the other hand, and in contrast to RIDD-tagged PTDH, complex formation of the tagged PAMO variants showed that the RIDD tag functions properly, when fused with PAMO, allowing dimer formation and self-assembly with the RIAD tag.

### 3.3. Activity and Stability Analyses

Next, the PTDH and PAMO activity of the obtained complexes were measured using phosphite and phenylacetone as substrates, respectively [27]. The PTDH–PAMO fusion enzyme was used as a control, together with the isolated enzymes (Table 2). Overall, all tested enzyme complexes and the fusion enzyme exhibit higher PAMO activities when compared with PAMO. RIDD-tagged PAMO showed a similar high activity when compared with PAMO fused to PTDH. PTDH activity in the complexes was similar to the activity observed for PTDH. Only RIAD-tagged PTDH in complex with RIDD-tagged PAMO showed a similar high PTDH activity as observed in PTDH when fused to PAMO. The hydrodynamic and kinetic analyses indicate that the RIDD and RIAD tags can be compatible with complex formation and enzyme activity. In fact, complex formation seems to boost enzyme activity of both PAMO and PTDH when compared with the native enzymes.

To establish whether the complex assembly had an effect on enzyme stability, the apparent melting temperatures of PAMO and PTDH were determined by using the ThermoFAD (for PAMO) or ThermoFluor (for PTDH) methods, respectively [27]. The thermostability for PAMO when complexed through RIAD–RIAD interactions was similar to the thermostability of native PAMO, and PAMO fused to PTDH (Table 2). On the contrary, the thermostability of PTDH in fusion with tags among assemblies and mixtures of enzymes varied and dropped drastically. Although PTDH from the fusion of PTDH–PAMO also experienced a decrease, the effect was milder. It seems that all tags had a negative effect on PTDH (dimeric and small enzyme), with a less negative effect of the RIDD tag than the other two tags (Table 2). This might indicate that longer and dimeric tags have a beneficial effect on the thermostability of the PTDH in a mixture of enzymes. Recent study has also suggested that secondary and tertiary structure combined with the length of the tag also have an impact on the function, especially the thermostability of the enzyme [28]. The results suggest that PTDH is not a very suitable enzyme for enforcing complex formation by using tags.

### 3.4. Biotransformations

All four constructs were tested for conversion of bicyclo[3.2.0]hept-2-en-6-one as a prototype BVMO substrate, as previously described [23] (Figure 2). This racemic bicyclic ketone was converted into normal and abnormal lactone in a similar ratio by all assemblies. Incubation times of 5 and 24 h were the minimum and maximum times chosen to observe the efficiency of substrate conversion based on the optimization by previous work using PTDH–PAMO [23]. In all cases, 59–68% of substrate was converted within 5 h of incubation (Figure 3, Appendix A). The highest conversion was obtained with the covalent PTDH–PAMO fusion, which converted 68% of the substrate in 5 h. After 24 h, the RIDD-tagged PAMO assemblies performed similarly to the PTDH–PAMO fusion (around 90% conversion). The RIDD-tagged PTDH assemblies only reached about 78% conversion.

This is in line with the observation that the RIDD-tagged PTDH did not form larger assemblies. The fact that the PAMO_I_-based assemblies perform similar to PTDH–PAMO shows that such assemblies provide a more efficient biocatalytic system because they allow a lower amount of PTDH. The conversion of the PTDH_A_–PAMO_I_ assembly was based on 5.0 µM tagged PAMO and 2.5 µM tagged PTDH, while the PTDH_A2_–PAMOI assembly conversion relied only on 5.0 µM tagged PAMO and 1.25 µM PTDH. This is in line with the fact that PAMO has a lower catalytic rate when compared with PTDH. The RIDD/RIAD system allows a better tuning of both enzyme concentrations, thereby effectively lowering the enzyme loading.

## 4. Conclusions

In this study, several PAMO–PTDH enzyme assemblies have been designed by using RIDD- and RIAD-tags. The respective RIDD- and RIAD-tagged enzymes were successfully produced in *E. coli* as soluble and active enzymes. Gel permeation experiments revealed that, of four theoretically possible assemblies, only two were actually formed: PTDH_A_–PAMO_I_ and PTDH_A2_–PAMO_I_. The impact of each tag on the catalytic activity and thermostability of enzymes–tag fusion was assessed by comparison with the native enzymes and the covalent PTDH–PAMO fusion. No drastic detrimental effects on activity of both enzymes were observed upon assembly. Moreover, no effect on the thermostability of PAMO was observed, while in some cases PTDH displayed a lower stability. Several RIDD/RIADD-mediated PAMO–PTDH assemblies were used for conversions. This revealed that the assemblies based on RIDD-tagged PAMO performed when compared with the covalently fused PTDH–PAMO. This means that the amount of required PTDH could be reduced by a factor of four. These results show that the dock-and-lock system of the RIDD/RIADD tag provide a method to design pre-defined assemblies of biocatalysts. Such enzyme complexes can be exploited for optimizing multi-enzyme biocatalytic conversions.

## Figures and Tables

**Figure 1 biomolecules-11-00905-f001:**
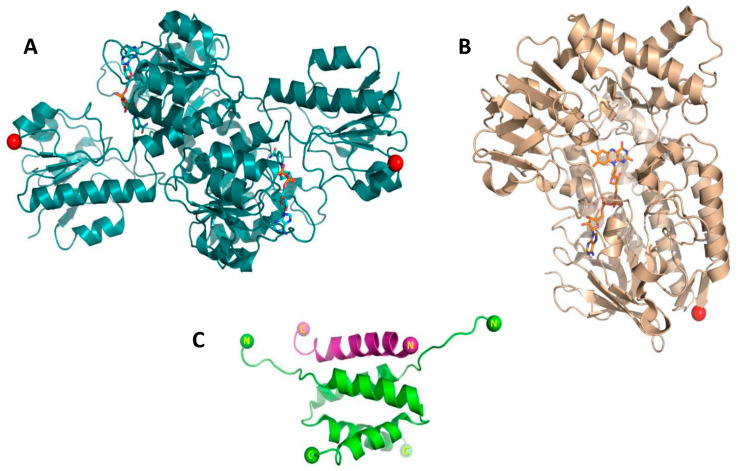
(**A**) Structure of dimeric PTDH, with NAD^+^ in cyan sticks; (**B**) Structure of monomeric PAMO with FAD in orange sticks. The C-termini are indicated with red spheres; (**C**) Structure of the docking and dimerization domain (RIDD, in green) of regulatory subunit of protein kinase A in complex with a peptide from an A-kinase anchoring protein (RIAD, in purple). The figures were prepared using PyMol, using PDB:2DRN, PDB:4E5P and PDB:1W4X, respectively.

**Figure 2 biomolecules-11-00905-f002:**
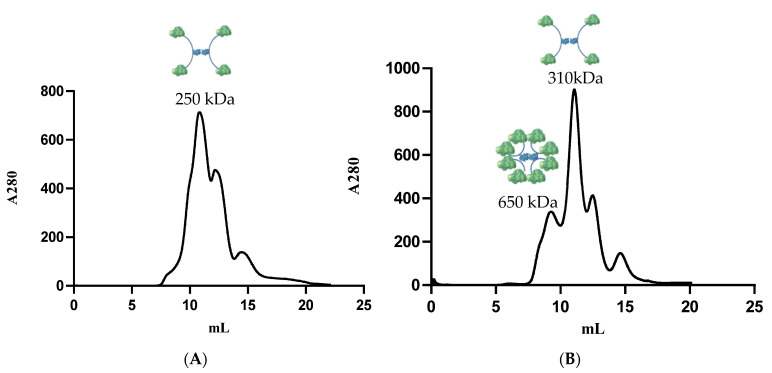
Gel filtration chromatograms of the enzyme mixtures. (**A**) mixture of PTDH_A_ and PAMO_I_; (**B**) mixture of PTDH_A2_ and PAMO_I_.

**Figure 3 biomolecules-11-00905-f003:**
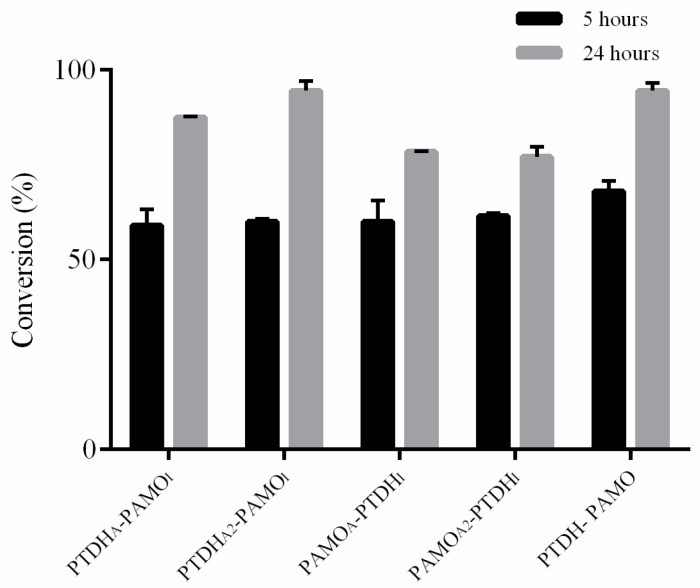
Biotransformation of 5.0 mM bicyclo[3.2.0]hept-2-en-6-one by variants of PAMO, PAMO_A_, PAMO_A2_, and PAMO_I_. Enzyme assemblies and fusion enzyme were composed of 5.0 µM of the respective PAMO variant and complemented with the respective PTDH concentration based on the used tags.

**Table 1 biomolecules-11-00905-t001:** Tagged enzyme variants.

Enzyme Variant ^a^	Linker	C-Terminus	Mw (kDa) ^b^
PAMO_A_	(GGGGS) × 3	RIAD	67
PAMO_A2_	(GGGGS) × 3	RIAD–RIAD	70
PAMO_I_	(GGGGS) × 3	RIDD	70
PTDH_A_	(GGGGS) × 3	RIAD	42
PTDH_A2_	(GGGGS) × 3	RIAD–RIAD	45
PTDH_I_	(GGGGS) × 3	RIDD	45

^a^ A: RIAD tag; A2: RIAD–RIAD tag; I: RIDD tag; ^b^ Molecular weight of each fusion enzyme was estimated using https://web.expasy.org/protparam/; accessed on 22 February 2020) based on the full sequence and confirmed with SDS-PAGE.

**Table 2 biomolecules-11-00905-t002:** Activity and stability of enzymes and enzyme complexes.

	PAMO		PTDH	
Enzyme	*k*_obs_ (s^−1^)	T_m_^app^ (°C) ^a^	*k*_obs_ (s^−1^)	T_m_^app^ (°C) ^b^
PTDH_A_–PAMO_I_	0.85 ± 0.01	59.0	4.18 ± 0.16	53.0
PTDH_A2_–PAMO_I_	0.86 ± 0.04	59.0	2.79 ± 0.15	51.5
PAMO_A_–PTDH_I_	1.08 ± 0.01	59.0	2.07 ± 0.02	56.5
PAMO_A2_–PTDH_I_	1.06 ± 0.08	59.5	2.12 ± 0.02	56.5
PTDH–PAMO	1.13 ± 0.01	60.0	4.65 ± 0.04	59.0
PAMO	0.58 ± 0.01	60.5	-	-
PTDH	-	-	2.42 ± 0.04	64.0

^a^ ThermoFAD; ^b^ Thermofluor. SYPRO orange (10× final concentration) was added to the sample.

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
