# Peer review of "Modular Assembly of Phosphite Dehydrogenase and Phenylacetone Monooxygenase for Tuning Cofactor Regeneration"

_biomolecules, 2021, doi:10.3390/biom11060905_

Round 1

Reviewer 1 Report

This is a descriptive study that uses a dock-and-lock system, based on the peptide-peptide interactions occurring between the regulatory subunit of protein kinase and the A-kinase anchor protein, to provide a physical support for the spatial approximation of two proteins that are meant to share metabolites. Several constructs of PTDH and PAMO are produced with those tags and the heterologous complexes produced are characterised in terms of number of protomers stablishing the complexes, thermostability, activity and efficiency in biotransformations. These parameters for the complexes formed are evaluated in the context of a properties of individual proteins and on a system fusing both proteins. Overall data indicate improvement of activity of some complexes regarding individual proteins, but not regarding the fused system. In addition, tagging of PTDH has a deleterious effect on its stability. The approach does not appear to particularly improve this as biotransformations system, because of PTDH, but might be of relevance for other proteins.

The data presentation and discussion are correct. Below I highlight some of the minor points that should be addressed:

Abstract: revise the sentence “In this study, we explored the use of the dimerization docking domain (RIDD) and anchoring domain (RIAD) system”. It is confusing because nothing is none about these domains yet.

It is not clear to me that “self-assembled PAMO-PTDH complexes performed slightly better when compared with the PTDH-PAMO fusion enzyme”. Make this sentence more realistic

Introduction and Methods

Page 2 Line 54. This is the first time PTDH is mentioned in introduction, provide full name.

Page 4. Line 126. RIAD-tagged enzyme concentration is given in mg, and twice molar excess of the second component is used. Would it be possible to provide these number in molar concentrations?

Page 4, line 134. “Fractions containing enzymes complexes were pooled and concentrated for further analysis using SDS-PAGE, ThermoFAD and ThermoFluor”. Something must be wrong in this sentence. Did you concentrate my using these methods or before using them?

Page 5. Line 157. “The mixture was extracted two times by mixing one volume of ethyl acetate including 0.025 mM internal standard methylester of benzoic acid for 60 s. Anhydrous magnesium sulphate to remove residual water”. Revise sentence, it is not clear.

Results

Page 6. Line 219. Replace “two protein” by “two proteins”.

Page 7. Line 254. Do you have any explanation for the repeat of RIAD being incompatible with multimer formation?

Page 8. Line 281. I think there is no Table 3. Table 2? In this context, do you thing longer linkers will be beneficial?

Page 9. Table 2. Error +-0.00??? better +.0.01

Page 10. Line 312. “This is in line with the fact that PAMO has a catalytic rate when compared with PTDH”. Something missing, the sentence those not read well.

Author Response

We thank the reveiwer for the comments, and used them to improve the quality of the manuscript. Below is our rebuttal. Changes in the manuscript are in yellow.

Reviewer #1

Abstract: revise the sentence “In this study, we explored the use of the dimerization docking domain (RIDD) and anchoring domain (RIAD) system”. It is confusing because nothing is none about these domains yet.

  • For clarity, we have rewritten this part of the abstract.

It is not clear to me that “self-assembled PAMO-PTDH complexes performed slightly better when compared with the PTDH-PAMO fusion enzyme”. Make this sentence more realistic

  • We thank the reviewer for the comment: we corrected the respective sentence.

Page 2 Line 54. This is the first time PTDH is mentioned in introduction, provide full name.

  • We corrected this.

Page 4. Line 126. RIAD-tagged enzyme concentration is given in mg, and twice molar excess of the second component is used. Would it be possible to provide these number in molar concentrations?

  • We added the concentration of the RIAD-tagged enzyme (20 µM).

Page 4, line 134. “Fractions containing enzymes complexes were pooled and concentrated for further analysis using SDS-PAGE, ThermoFAD and ThermoFluor”. Something must be wrong in this sentence. Did you concentrate my using these methods or before using them?

  • We corrected this.

Page 5. Line 157. “The mixture was extracted two times by mixing one volume of ethyl acetate including 0.025 mM internal standard methylester of benzoic acid for 60 s. Anhydrous magnesium sulphate to remove residual water”. Revise sentence, it is not clear.

  • We corrected this.

Page 6. Line 219. Replace “two protein” by “two proteins”.

  • We corrected this.

Page 7. Line 254. Do you have any explanation for the repeat of RIAD being incompatible with multimer formation?

  • We do not have a good explanation and could only speculate on structural incompatibility. Yet, from the structure of PTDH there is no clear basis for such suggestion. Therefore, we prefer to refrain from suggestion a reason for this and merely mention this as observation.

Page 8. Line 281. I think there is no Table 3. Table 2? In this context, do you thing longer linkers will be beneficial?

  • This should be indeed Table 2 (we once had two tables, but these have been merged).

Page 9. Table 2. Error +-0.00??? better +.0.01

  • We have changed it according to the suggestion: the error was really small.

Page 10. Line 312. “This is in line with the fact that PAMO has a catalytic rate when compared with PTDH”. Something missing, the sentence those not read well.

  • We corrected this.

Reviewer 2 Report

see PDF file

Author Response

We thank the reviewer for the positive evaluation and the indicated minor points. We have used the comments to improve the quality of the manuscript. A rebuttal is below and changes in the manuscript are in yellow.

Minor points:

Terms such as “protein expression” are strictly speaking incorrect (only genes are expressed), consider using “heterologous production” or similar terms

> We have corrected this accordingly.

  1. 1, line 41: consider rephrasing “by fusing the two respective enzymes at DNA level” to, e.g., with “by fusing the respective genes encoding these enzymes”

> We have corrected this accordingly.

  1. 1, lines 43-45: a citation should be added following that statement

> We have added a relevant reference.

  1. 2, line 68: correct to “loss of enzyme activity”

> We have corrected.

  1. 5, line 182: “tagged” used twice in sentence, “the tagged enzymes” could e.g., simply be replaced by “they”.

> We have corrected this accordingly.

  1. 6, line 209 ff.: Not entirely convincing that a 364 kDa complex elutes at a time corresponding to a 250 kDa complex. Complex composition could be checked by other methods too, e.g. native PAGE or by quantifying the relative protein amounts and stoichiometric composition based on the spot intensities of each component on a SDS-gel (while also taking into account the respective molecular masses).

> We agree that the difference in size is significant. We did confirm by SDS-PAGE both proteins being present. The complexes will not be globular in shape and may rearrange during the size exclusion step, which may translate into aberrant behavior in gel permeation experiments. This is why we indicate that the hydrodynamic behavior may play a role in this apparent difference.

p.6, line 219: correct to “proteins”

> We have corrected this.

  1. 7, line 222: consider rephrasing, e.g., to: “…and corresponds to a complex with half the size…”

> We have corrected this accordingly.

Table 2, line 290: correct to “SYPRO orange (10x final concentration) was added to the sample”

> We have corrected this accordingly.

  1. 10, line 312: word missing, should probably read “This is in line with the fact that PAMO has a lower catalytic rate….”

> We have corrected this.

p.10, line 322: correct “by in comparison with” to “by comparison with”

> We have corrected this.